# Seaweed Habitats on the Shore: Characterization through Hyperspectral UAV Imagery and Field Sampling

Wendy Diruit [1,*], Anthony Le Bris [2], Touria Bajjouk [3], Sophie Richier [2], Mathieu Helias [1], Thomas Burel [1], Marc Lennon [4], Alexandre Guyot [4] and Erwan Ar Gall [1]

1    Univ Brest, CNRS, IRD, Ifremer, LEMAR, 29280 Plouzané, France;
     mathieu.helias@etudiant.univ-brest.fr (M.H.); thomas.burel@univ-brest.fr (T.B.);
     erwan.argall@univ-brest.fr (E.A.G.)
2    Centre d'Etude et de Valorisation des Algues (CEVA), 22195 Pleubian, France; anthony.lebris@ceva.fr (A.L.B.);
     sophie.richier@ceva.fr (S.R.)
3    Ifremer, Dynamiques des Ecosystèmes Côtiers (DYNECO)/Laboratoire d'Ecologie Benthique
     Côtière (LEBCO), 29280 Plouzané, France; touria.bajjouk@ifremer.fr
4    Hytech-Imaging, 115 Rue Claude Chappe, 29280 Plouzané, France; marc.lennon@hytech-imaging.fr (M.L.);
     alexandre.guyot@hytech-imaging.fr (A.G.)
*    Correspondence: wendy.diruit@univ-brest.fr

**Abstract:** Intertidal macroalgal habitats are major components of temperate coastal ecosystems. Their distribution was studied using field sampling and hyperspectral remote mapping on a rocky shore of Porspoder (western Brittany, France). Covers of both dominating macroalgae and the sessile fauna were characterized in situ at low tide in 24 sampling spots, according to four bathymetric levels. A zone of ca. 17,000 m$^2$ was characterized using a drone equipped with a hyperspectral camera. Macroalgae were identified by image processing using two classification methods to assess the representativeness of spectral classes. Finally, a comparison of the remote imaging data to the field sampling data was conducted. Seven seaweed classes were distinguished by hyperspectral pictures, including five different species of Fucales. The maximum likelihood (MLC) and spectral angle mapper (SAM) were both trained using image-derived spectra. MLC was more accurate to classify the main dominating species (Overall Accuracy (OA) 95.1%) than SAM (OA 87.9%) at a site scale. However, at sampling points scale, the results depend on the bathymetric level. This study evidenced the efficiency and accuracy of hyperspectral remote sensing to evaluate the distribution of dominating intertidal seaweed species and the potential for a combined field/remote approach to assess the ecological state of macroalgal communities.

**Keywords:** seaweeds; hyperspectral; UAVs; intertidal ecology; rocky shores; supervised classification; vegetation cover

## 1. Introduction

The intertidal zone hosts considerable diversity together with a great abundance of benthic organisms [1,2] and has long been monitored as a control ecosystem in ecological processes. Seaweeds are the major component of flora on temperate rocky shores, where they can commonly form extensive canopies, structuring macroalgal communities comparable to terrestrial forest systems in their arrangement [3,4]. Seaweed species are vertically distributed on the shore according to several abiotic factors such as desiccation, hydrodynamics, light and salinity, themselves largely influenced by tide oscillations [5,6]. Temperate rocky shores are globally dominated by fucoids (i.e., large Phaeophyceae from the order Fucales), from high to low levels of the shore and by kelps (i.e., large Phaeophyceae from the order Laminariales *sensu lato*) in the lower intertidal fringe and the subtidal area [7]. Along the north east Atlantic coastline, up to six successive macroalgal communities may

be found [8,9], which can be reduced to 2–5 depending on the geographical area, the substratum or the hydrodynamic conditions [10].

Brittany is a long-term monitored area for macroalgal diversity (approximatively 650 species [11]) and resources (e.g., Benthic Network research program since 2005). These characteristics are examples of a prime area to fully describe seaweed-dominated habitats through remote sensing. Therefore, remote sensing for macroalgal covers has undergone early development since the 1960s [12–17].

Seaweed communities have been recognized as a quality element for the classification of coastal water bodies as part of the European Water Framework Directory (WFD, 2000/60/EC; E.C., 2000 [18]) and several metrics based on the good ecological state of macroalgal communities have been developed along the European coasts [6,9,19–24]. On rocky shores, the occurrence and abundance of vegetation can easily be estimated visually through the cover-abundance scale, or percentage-cover indices, without damaging the habitat [5]. Even if these estimations are easy to implement, they may be time consuming and some locations remain difficult to reach. In this context, using remote sensing imagery for spatialization is an interesting alternative to a site-specific scale [25,26], and could help survey shifting ecosystems [27].

Both multispectral and hyperspectral imagery are routinely used on terrestrial vegetation, for instance, to estimate crop yields [28–31]. By contrast with other plants, seaweeds have a larger phylum-specific diversity of pigments, which can be discriminated by analyzing spectral characteristics at different wavelengths [32]. Pigment diversity in algae contributed to the early development of seaweed detection through airborne remote sensing [33]. Later, mapping of macroalgal communities was processed using satellite imagery (IKONOS, SPOT, Sentinel-2), with scale refining depending on the sharpness of the sensors aboard [34–37], and promoted combined airborne/ground spectra acquisition for macroalgal mapping. Another powerful tool to study coastal environments is the use of free-access satellite images, which could help to produce extensive habitat mapping, in order to observe natural variations in habitats overtime [38].

These methods enable the collection of homogeneous data over broad spatial scales but are inaccurate when applied to heterogeneous habitats, varying at a centimeter in scale [39]. Such approaches are complexified in coastal areas due to tidal variations and highly mosaic environments [40]. Furthermore, data acquisition is generally altered by the occurrence of a water layer [41] and often disturbed by atmospheric conditions (noticeably, cloud cover and light reflection). The development and easy access to both unmanned aerial vehicles (UAVs) and hyperspectral sensors further promoted the remote mapping and characterization of intertidal habitats [42–44]. Since the 1970s, automated methods (i.e., classification algorithms) have been developed to classify multi-/hyper-spectral images [45–47]. The present work focuses on an easy habitat classification through high spatial resolution pictures, obtained by a UAV and by applying commonly used algorithms. To characterize seaweed-dominated habitats, maximum likelihood (MLC) is currently the most widely used method of supervised classifications [48], along with the spectral angle mapper (SAM) [49–52].

To date, there are still few studies comparing both mapping intertidal seaweed using multispectral [38,42,53] or hyperspectral sensors on UAVs, and an accurate spatial resolution (less than 5 cm). Indeed, the majority of studies focus on kelp beds at lower resolution (spatial and/or spectral) [54–57]. Rossiter et al. (2020) [49,58] successfully classified shores using both multispectral and hyperspectral sensors focusing on the Fucales *Ascophyllum nodosum*, but did not compare sensor data with macroalgal in situ covers.

To fill these gaps existing between remote sensing and field sampling, a two-way approach was conducted: on the one hand, in situ sampling of macroalgal communities, and on the other hand, hyperspectral UAV imagery acquisition and automated classifications. In that prospect, several objectives were defined:

1. Distinguishing macroalgae from seawater, substratum and associated non-algal organisms based on classification results from hyperspectral imagery.

2. Using hyperspectral data to discriminate the main species of fucoids from green and red macroalgae.

3. Testing the accuracy of supervised classification algorithms.

4. Comparing field and remotely estimated cover-abundance data.

The working hypothesis of this study is that the two classifications, obtained from hyperspectral images, would yield similar results between the two, successfully differentiating macroalgae and would correspond to those obtained in the field. The present experiment was performed on a seaweed-dominated shore of western Brittany to test the complementarity between both approaches and to study the distribution of seaweed habitats. The aim of the study is to evaluate the correspondence between the distribution of species in macroalgal habitats obtained by both in situ sampling on the shore and hyperspectral imagery by a UAV.

## 2. Materials and Methods

### 2.1. Studied Site and Communities

The study was performed on the coasts of north-west Brittany, on the site of Porspoder (48°28.88′ N/4°46.29′ W) (Figure 1). The site is about 230 m long and 100 m wide, with a maximal tidal range of 8.15 m and mainly exhibiting dense macroalgal canopies with some pools, boulder fields and bedrock. The six macroalgal communities typically found in the north-east Atlantic were vertically distributed on the shore [8] and were grouped into 4 bathymetric levels for the study. These levels correspond to either single or mixed communities named by the dominating Fucales and Laminariales: (1) *Pelvetia canaliculata* plus *Fucus spiralis* communities, (2) *Ascophyllum nodosum*/*Fucus vesiculosus* community, (3) *Fucus serratus* community and (4) *Himanthalia elongata*/*Bifurcaria bifurcata* plus *Laminaria digitata* communities. These levels are referred to, respectively, as Pc-Fspi, An, Fser and He-Ld hereafter.

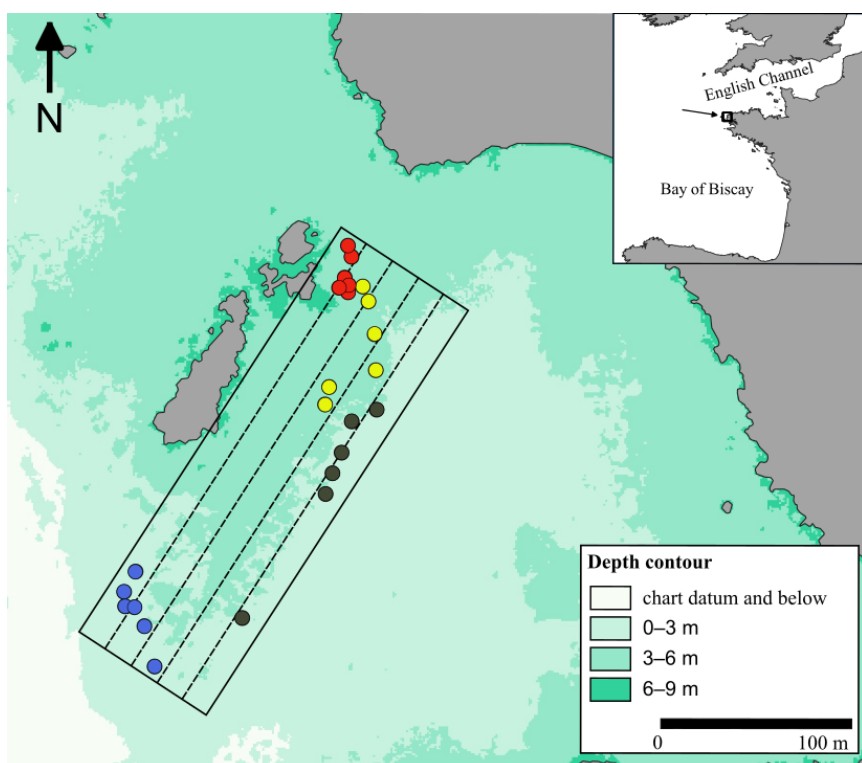

**Figure 1.** Study site of Porspoder (Brittany, France: 48°28.88′ N/4°46.29′ W) showing the 24 in situ sampling spots surveyed during the study. The color of the circles indicates the intertidal level considered: red circles, *P. canaliculata—F. spiralis*; yellow circles, *A. nodosum*; black circles, *F. serratus*; blue circles, *H. elongata*. The dotted lines correspond to the UAV flight lines.

### 2.2. Sampling Method

Field sampling was conducted in Spring 2021 (28 April to 1 June). A total of 24 sampling spots (i.e., 6 for each of the 4 levels) were monitored at low tide. The sampling spots were referenced using pictures and GPS positioning (Garmin GPS 73, ±3 m). The sampling protocol followed the methodology described in Burel et al. (2019b) [59]. A mobile plastic grid structure of 1.65 m × 1.65 m divided into 25 quadrats of 33 cm × 33 cm was used to delimit each sampling spot. Covers of benthic fauna, flora and bare rock were estimated visually on the entire surface delimited by the plastic structure from 0 to 100 percent, with a 5 percent pace. That approach, known as 'undisturbed sampling', describes the distribution of the main groups of benthic organisms plus the substratum during emersion.

### 2.3. Remote Sensing Acquisition

Acquisitions were made by Hytech-Imaging (Plouzané, Brittany, France) using a NEO HysPex Mjolnir V-1240 sensor (Oslo, Norway) (Table 1). The sensor was set on an octocopter UAV based on Gryphon Dynamics X8 architecture (Figure 2), with a gStabi H16 stabilization, containing an Applanix APX15 inertial unit with an L1/L2 GPS receiver and a GPS L1/L2 Tallysman enabling geolocation. The UAV and the central acquisition unit of the sensor were remotely controlled by a radio link.

**Table 1.** Characteristics of the hyperspectral visible near infrared (VNIR) Mjolnir_V-1240 sensor. FOV = field of view.

| Spectral Range | Spatial Pixels | Spectral Resolution | Spectral Sampling | Number of Bands | FOV Across Track | iFOV Across/ 3Along Track | Coding |
|---|---|---|---|---|---|---|---|
| 0.4–1 μm | 1240 | 4.5 nm | 3 nm | 200 | 20° | 0.27/0.27 mrad | 12 bits |

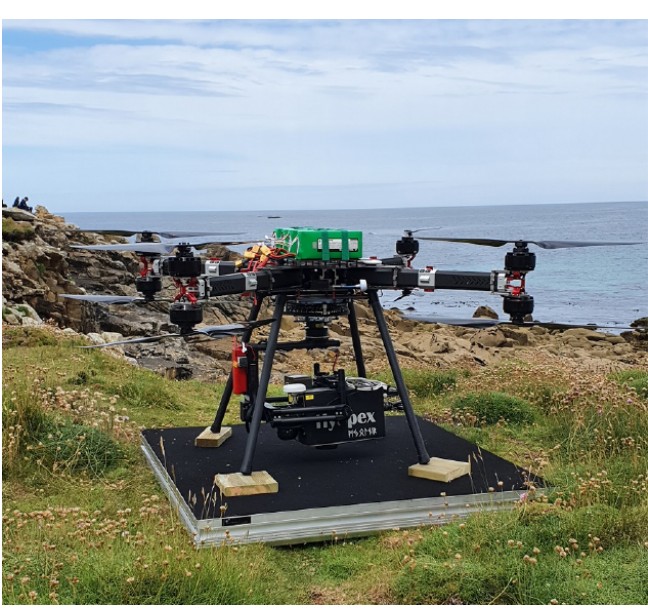

**Figure 2.** UAV octocopter used for the acquisitions.

Acquisitions were performed on the 24 June 2021 at a 64 m height to obtain a resolution of 2 cm (Table 2). To perform the image acquisition, two technicians were involved to pilot the UAV and to operate the hyperspectral sensor. The acquisition lasted about 30 min. The flight plan was designed to cover a subsection of the site of Porspoder, including all of the field sampling spots (Figure 1).

**Table 2.** Parameters of the aerial survey.

| Flight Altitude | Ground Sampling Distance | Swath | Mapped Area | Viewing Angle | Flight Lines |
|:---:|:---:|:---:|:---:|:---:|:---:|
| 64 m | 2 cm | 23 m | 1.76 ha | 20° | 4 |

Images (Figure 3) were collected between 09h28 and 09h47 UTC at low tide (tidal coefficient 92 corresponding to tidal range of 6.1 m). During the acquisitions, light was diffused due to cloud cover.

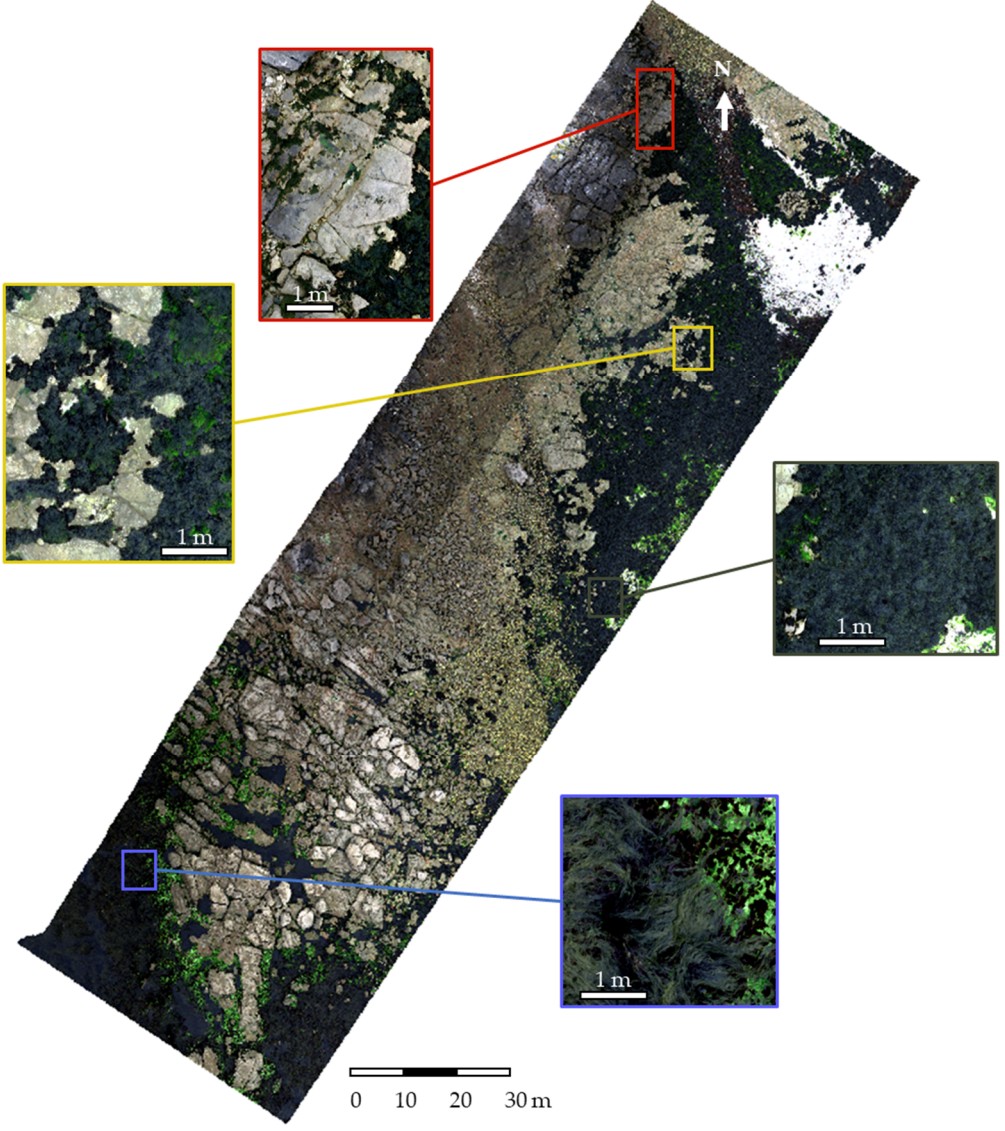

**Figure 3.** Porspoder orthophoto (RGB) obtained during the flight on the 24 June 2021. Detailed sections of the color image illustrating the different bathymetric levels on the shore are represented: Pc-Fspi (red square), An (yellow square), Fser (black square) and He-Ld (blue square).

*2.4. Pre-Processing*

To obtain a georeferenced image in spectral radiance (W·m$^{-2}$·sr$^{-1}$·µm$^{-1}$), the hyperspectral image was processed from raw data (level 0) to a radiometrically and geometrically calibrated image (level 1c) using the HYPIP (HYPperspectral Image Preprocessing) chain of Hytech-imaging that includes ATCOR/PARGE software applications (ReSe Applications, Wil, Switzerland). To calculate the surface reflectance, atmospheric corrections were

performed in a two-step process: first, using the ATCOR-4 software, and then empirically adjusting each spectrum. To adjust each spectrum, coefficients of gain and bias were calculated per spectral band, by linear regression between surface reflectance data and the reflectance signature. This reflectance signature was obtained by positioning pre-calibrated targets (tarps) near the area of interest overflown during the survey.

### 2.5. Data Classification

For this study, supervised classifications were performed, where categories (classes) correspond to spectral signatures defined by the user. A class contains a characteristic spectral signature for each dominating fucoid species, macroalgal group or an abiotic component and corresponds to homogeneous regions delineated on the UAV image. The software then assigns each pixel of the image into a cover type to which its signature is most comparable [60]. The supervised classifications were performed after defining regions of interests (ROIs) which are training data. ROIs were created for each class using 'ROI tool' in ENVI version 5.6.1 (Exelis Visual Information Solutions, Boulder, CO, USA) by manually circling pixel areas on the image. More than one training ROI were usually used to represent a particular class (ROIs = multiple polygons) (Table 3). The number of polygons and pixels per class depend on the surface occupied by each species. For example, covers of *P. canaliculata* and *F. spiralis* are low compared to those of *F. serratus* or *H. elongata*, which represent more homogeneous and larger classes. Classes were selected in agreement with the hyperspectral image and pictures taken during field sampling.

**Table 3.** Number of ROIs and pixels for each class.

| Class | Number of ROIs | Number of Pixels |
|---|---|---|
| *P. canaliculata* | 76 | 29,899 |
| *F. spiralis* | 10 | 551 |
| *A. nodosum* | 233 | 334,002 |
| *F. serratus* | 227 | 894,910 |
| *H. elongata* | 145 | 353,825 |
| Green | 482 | 73,592 |
| Red | 509 | 41,808 |
| Substratum | 408 | 1,834,496 |
| Water | 235 | 1,073,044 |

Nine classes were thus defined for the site of Porspoder (Figure 4), including five classes of dominating Fucales ('*Pelvetia canaliculata*', '*Fucus spiralis*', '*Ascophyllum nodosum*', '*Fucus serratus*' and '*Himanthalia elongata*'), and two classes related to green and red seaweeds (respectively, 'Green' and 'Red') were created. A 'Substratum' class was defined grouping bedrock, boulders, gravel and sand, and, finally, a 'Water' class was also created, gathering immerged parts of the shore (pools or subtidal zone). The 'Substratum' and 'Water' classes were classified in the same way as the other classes and have subsequently been removed from the maps to improve their clarity and interpretation. Due to the complexity of accurately identifying benthic fauna on the UAV image, no appropriate class was created, and these data were grouped together as the 'Substratum' class.

Training data (i.e., ROIs mean spectra) were checked for class separability using the Jeffries–Matusita distance [61]. The values of the resulting output between each pair of classes ranged between 0 and 2, with values greater than 1.9 indicating almost perfect separability between them [46]. A large class separability indicates that accurate training areas have been selected, whereas values approaching zero suggest either the need for more training areas or classes that are inherently similar in their spectral properties.

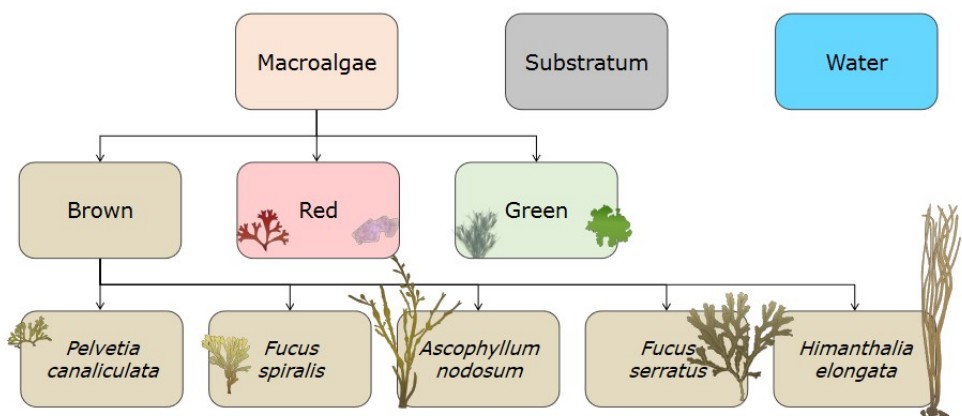

**Figure 4.** Hierarchical tree of decision to make classes, inspired from Congalton et al. (1999) [62].

Two supervised classification methods were performed to test the representativeness of the spectral classes running the software ENVI version 5.6.1 (Exelis Visual Information Solutions, Boulder, CO, USA), i.e., the algorithms maximum likelihood classification (MLC) and spectral angle mapper (SAM).

MLC calculates the probability that an individual pixel belongs to a specific class and is based on an estimated probability density function derived from the defined reference classes [63]. MLC is a popular classifier [64]. The use of spectral profiles by this method requires ROIs based on multiple pixels. Following this method, the classification is based on the selection of the most representative spectral profiles in ROIs of the same class upon different flight lines. The MLC classifier assumes a Gaussian distribution for each input training class [65] and it can be expressed by the following equation:

$$g_i(x) = \ln p(\omega_i) - \frac{1}{2} \ln | \sum_i | - \frac{1}{2}(x - m_i)t \sum_i^{-1}(x - m_i) \tag{1}$$

where $i$ is a given spectral class, $x$ equals n-dimensional data, $p(\omega_i)$ is the probability that class $\omega_i$ occurs in the image and it is assumed the same for all classes, $| \sum_i |$ is the

determinant of the covariance matrix of the data in class $\omega_i$, $\sum_i^{-1}$ is the inverse matrix and $m_i$

is the mean vector. The advantage of MLC as a parametric classifier is that it considers the variance-covariance within the class distributions and, for normally distributed data, MLC performs better than the other known parametric classifiers [66]. However, for data with a non-normal distribution, the results may be unsatisfactory.

SAM identifies the spectral similarity between two spectra collected from an image or distributed from a spectral library [67]. The resulting classification is rather based on the angular orientations of spectral vectors [29]. Similarities within pairs of spectra (reference and classification) can be compared regardless of differences in brightness, and the pairs are treated as vectors in an n-dimensional space [68]. SAM is expressed by the following equation, taken from Kruse et al. 1993 [67]:

$$\alpha = cos^{-1} \left[ \frac{\sum_{i=1}^{nb} t_i r_i}{\left(\sum_{i=1}^{nb} t_i r_i\right)^{\frac{1}{2}} \left(\sum_{i=1}^{nb} t_i r_i\right)^{\frac{1}{2}}} \right] \tag{2}$$

where $t$ is the spectra for a pixel, $r$ is for the reference spectrum pixel, $\alpha$ is the spectral angle between $t$ an $r$ (measured in radians or degrees) and $n$ is the number of bands.

The average spectral reflectance curves from the ROIs were extracted since SAM requires endmember spectra. If two ROIs were identical, they were averaged in order to obtain one curve with the maximum possible data. The use of spectra derived directly from

the image is usually better than using ground or library spectra due to better inclusions of errors related to atmospheric corrections, calibration and effects of sensor responses [29].

For both classifications (SAM and MLC), no detection threshold was selected, so that all pixels could be classified.

### 2.6. Data Analysis

Accuracy assessment for classification was checked using ground truth (or reference) ROIs based on the same method as the training data [63,69]. These polygons were independent of the training ROIs and their number represented one third of training ROIs. The accuracy assessment tool was used to create the confusion matrix and derive quantitative measures of accuracy (i.e., kappa coefficient, overall accuracy, user/producer accuracy, errors of commission/omission) using ENVI version 5.6.1 (Exelis Visual Information Solutions, Boulder, CO, USA). User accuracy is the probability of correct class assignment, calculated by dividing the number of correctly classified pixels by the total number of pixels in the class, and producer accuracy is the correctly classified reference pixels, calculated by dividing the number of correctly classified pixels by the total number of pixels that should be in a class.

Each grid structure was replaced using 'Advanced Digitizing toolbar' ('Move Feature' and 'Rotate Feature' options) on Qgis. Corresponding polygons were accurately positioned using pictures taken during the field sampling, in order to decrease the potential GPS error and to compare the exact same position.

To compare in situ data and classification data, vectors of the grid structure were replaced on the Porspoder image using the 'Vector to ROI' tool, and the percentage of pixels for each class in ROIs was extracted with the ROI statistics tool on ENVI.

Statistical analyses were conducted using the R environment [70]. Normality and homoscedasticity were first tested on each biological and classification variable, corresponding to seaweed species and substratum covers, with Shapiro–Wilk and F Test/Levene tests, respectively. These tests then determined what analyses were the most suitable (parametric or not). In order to represent the distribution of the replicates described by the three approaches, a distance-based redundancy analysis (db-RDA) was constructed, based on the method described by Escobar-Briones et al. (2008) [71]. Values for each class (apart from 'Water') were first converted into a distance matrix by calculating the Hellinger distance for each class in the whole dataset. Then, a principal coordinates analysis (PCoA) was performed on this matrix. The PCoA allows to convert the distance between items (distance matrix) into a map-based visualization (each item is assigned a location in a low-dimensional space, materialized by its eigenvector) in order to better understand the relation between each object. All the PCoA eigenvectors were used as input into an RDA in order to build the db-RDA. The db-RDA represents on a single plot the position of the different replicates using the PCoA eigenvalues, as well as the species (classes) and the explanatory variables (level and method).

To compare more precisely the cover of each of the classes for the three methods and for each bathymetric level, Kruskal–Wallis tests (non-parametric) were performed followed by a post-hoc Dunn test to identify variables that were statistically different.

## 3. Results

### 3.1. In Situ Vegetation Cover

Large discrepancies were observed in the covers between bathymetric levels, the lowest levels being characterized by a dominance of seaweeds, whereas bare rock showed a large occurrence in the upper level. Covers of macroalgal classes differed between the four levels (Figure 5). The Pc-Fspi level, corresponding to the upper shore (5.2–6.1 m above chart datum (CD)), was lightly vegetalized, with bare rock occupying 55% of the surface. The level was dominated by the Fucales *P. canaliculata* for about 32.5%. The remaining covers were well distributed between *F. spiralis* and red seaweeds (5% each), whereas benthic fauna (barnacles and limpets) corresponded to a cover of 2.5%.

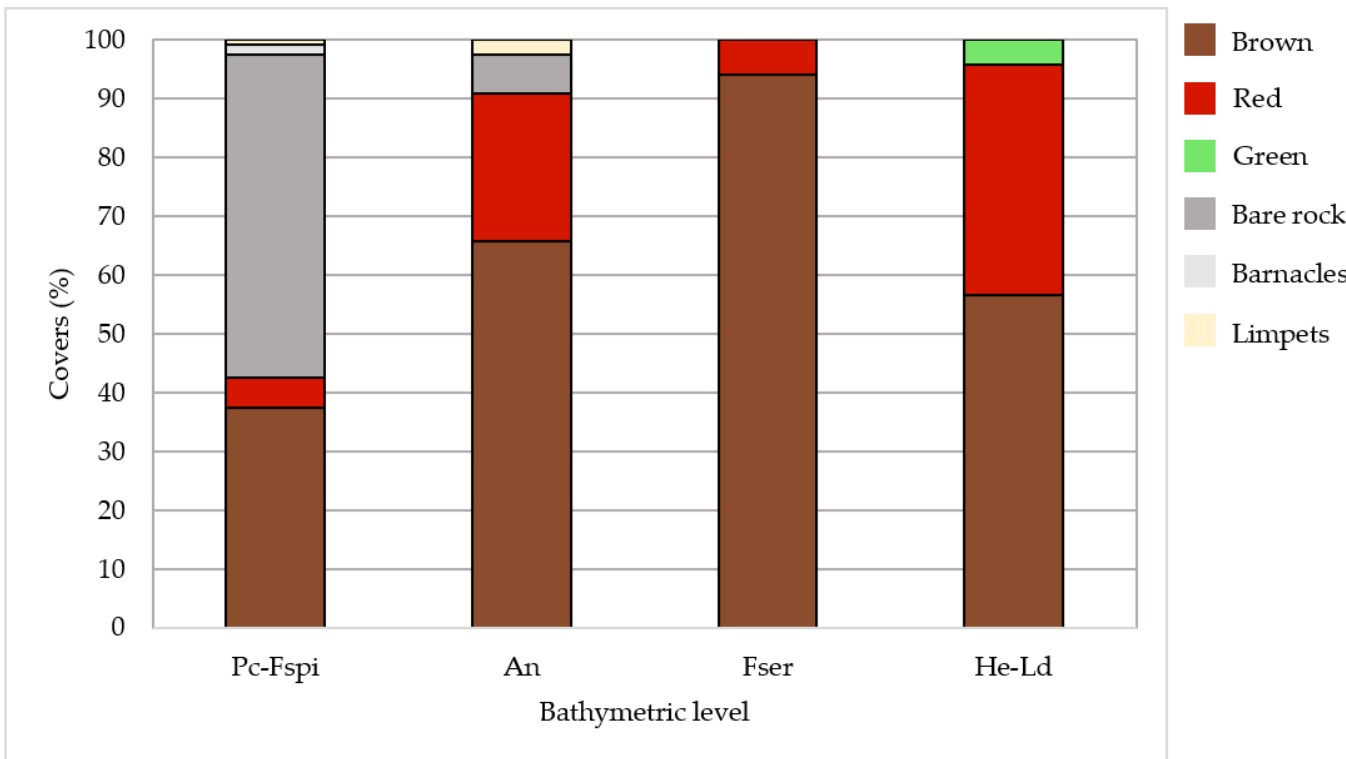

**Figure 5.** Average covers of macroalgal groups and sessile fauna, and percentage of bare rock observed in situ at each bathymetric level. Covers are given in percentages. Fucoids and other brown species are grouped in the 'Brown' class, and erect and crustose red algae are grouped in the 'Red' class.

The An level (middle shore, 3.4–4.4 m above CD) was largely dominated by the Fucales *A. nodosum* (60%) and *F. serratus* (5.9%). Red seaweeds then covered about 25% of the surface (22.5% erect and 2.5% crustose). Bare rock and limpets completed the remaining surface (6.7% and 2.5%, respectively).

In the Fser level (lower shore, 3.1–2.3 m above CD), macroalgal covers became conspicuously dominant compared to bare rock and sessile fauna. Indeed, the cover of *F. serratus* was close to 100% (94.2%), while the rest corresponded to erect red algae (5.8%).

In the He-Ld level (2.8–1.6 m ab. CD), the distribution between macroalgal groups was equilibrated, with a co-dominance of *H. elongata* (39.2%) and erect red seaweeds (36.7%). The Laminariales *L. digitata* also presented large covers (17.5%), and in addition, there were little covers of crustose red and green seaweeds (2.5% and 4.2%, respectively).

Thus, An, Fser and He-Ld had a higher cover of Phaeophyceae (more than one half) compared to the other macroalgal groups of species (65.8%, 94.2% and 56.7% of cover, respectively). By contrast, Pc-Fspi showed only a bit more than one third of cover by Phaeophyceae (37.5%).

*3.2. Classifications Results*

3.2.1. MLC Results

The results from the class separability test of image-derived spectra showed that all of the class pairs had values greater than 1.90, indicating globally a good class separation (Figure 6) [72].

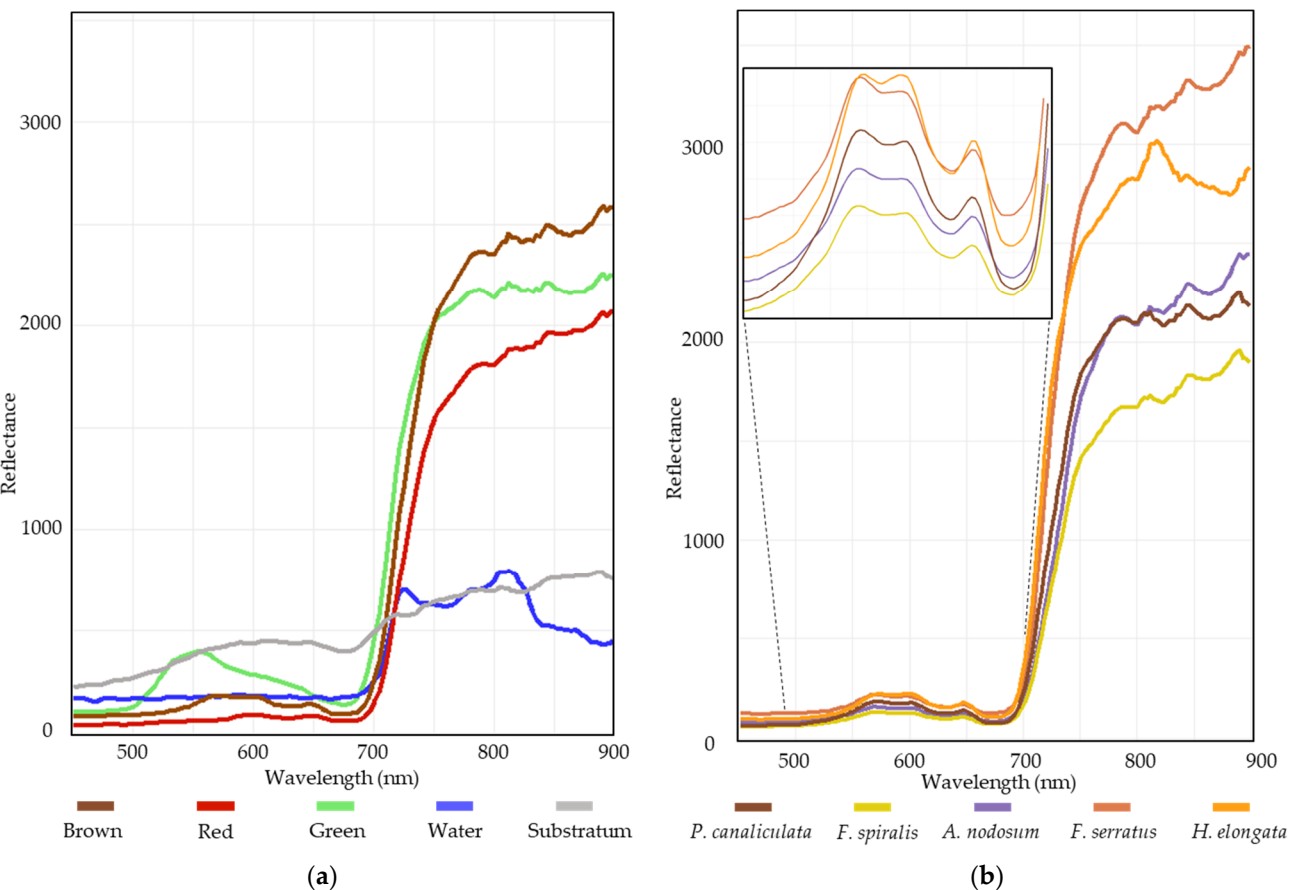

**Figure 6.** Average reflectance of the different spectral classes between 450 and 900 nm: (**a**) macroalgal groups, water and substratum mean reflectance; (**b**) detailed spectra of each Fucales.

The MLC classifier, trained using image-derived spectra, revealed a dense cover of intertidal Fucales (26.9% of the site) (Figure 7). The overall classification accuracy for the MLC was 95.1% and the kappa coefficient was 0.93. The four bathymetric/vegetation levels appeared clearly, forming four distinctive bands. The Pc-Fspi level (upper shore) was dominated by a thin band of both *P. caniculata* and *F. spiralis* (1.3% and 0.1% of total pixels, respectively). The An and Fser levels (mid-shore) were dominated by a large band of *A. nodosum* and of *F. serratus* (5.6% and 6.8% of total pixels, respectively). The He-Ld level (lower shore) was characterized by the important development of *H. elongata* (9.7% of total pixels) and a cover of red macroalgae greater than in higher levels (1.7% for all of the site). Green algae were mainly present in the lower shore (1.7% of total pixels). The 'Substratum' and 'Water' classes represented the majority of the site (51.8% and 21.3% of total pixels for the site, respectively). The macroalgal classes '*A. nodosum*', '*F. serratus*' and '*H. elongata*' showed the highest producer/user accuracies (Table 4). There were some misclassifications between the Fucales '*A. nodosum*' and '*F. serratus*' (1.56%), and between '*A. nodosum*' and '*P. caniculata*' (7.04%). The lowest producer/user accuracy was for '*F. spiralis,*' with some misclassifications between '*F. spiralis*' and '*P. caniculata*' (44.13%) and between '*F. spiralis*' and '*A. nodosum*' (14.81%). 'Green' and 'Red' algae were also well classified (96.92% and 90.54%, respectively) but there were some misclassifications between 'Red' algae and the Fucales '*P. caniculata*' (3.04%) and '*H. elongata*' (2.06%).

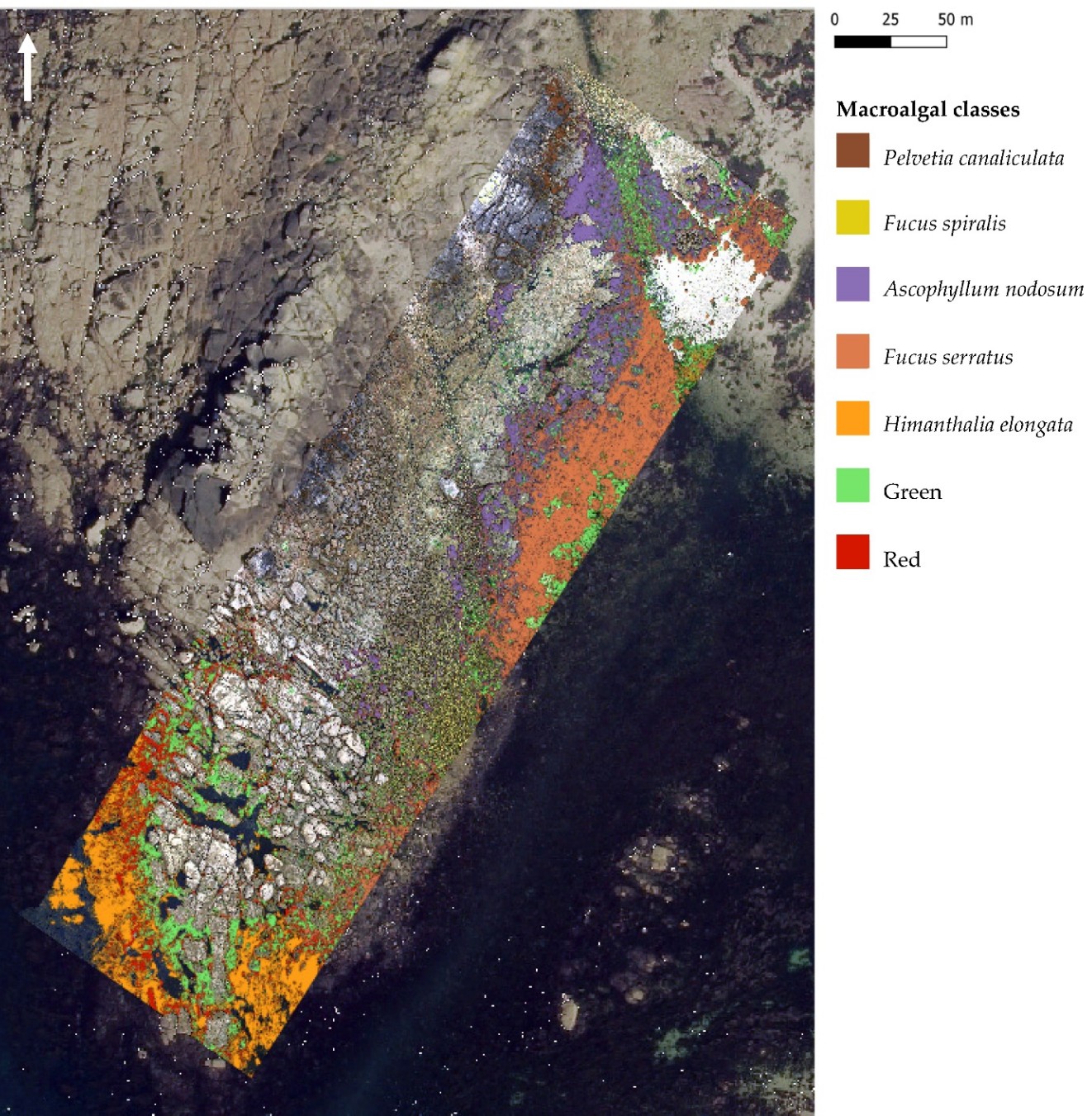

**Figure 7.** Maximum likelihood classification (MLC), trained using image-derived spectra resulting from the hyperspectral UAV survey at Porspoder. Seven macroalgal cover classes are displayed over the UAV RGB imagery and an orthophotography (Mégalis Bretagne et collectivités territoriales bretonnes—2015), giving an overview of the site. The 'Substratum' and 'Water' classes are not represented on the map. Class codes 'Green' and 'Red' represent grouped green and red macroalgal species, respectively.

**Table 4.** Maximum likelihood classification (MLC) confusion matrix, calculated, using ENVI 5.6.1, by comparing pixels of known class locations to those predicted by the classification workflow for each of the nine cover classes. Results are displayed as percentages of pixels assigned, correctly or incorrectly, to each class. User/producer accuracies (User Acc. and Prod. Acc., respectively) are also presented.

| Class | P. canaliculata | F. spiralis | A. nodosum | F. serratus | H. elongata | Green | Red | Substratum | Water | Total | User Acc. |
|---|---|---|---|---|---|---|---|---|---|---|---|
| Unclassified | 0 | 0 | 0 | 0 | 0 | 0 | 0 | 0 | 0 | 0 | - |
| P. canaliculata | 97.82 | 44.13 | 7.04 | 0.01 | 0.02 | 0 | 3.04 | 0.90 | 0.03 | 1.28 | 26.04 |
| F. spiralis | 0.13 | 39.00 | 0.23 | 0 | 0.07 | 0 | 0.85 | 0.09 | 0.24 | 0.14 | 6.23 |
| A. nodosum | 0.79 | 14.81 | 89.32 | 5.59 | 0.03 | 0.27 | 0.78 | 0.04 | 0.02 | 5.63 | 91.93 |
| F. serratus | 0.01 | 0 | 1.56 | 91.71 | 0.03 | 0.01 | 0.04 | 0 | 0 | 6.80 | 98.61 |
| H. elongata | 0.09 | 0.15 | 0.08 | 0.38 | 93.53 | 0.18 | 2.60 | 0 | 4.00 | 9.69 | 90.72 |
| Green | 0.17 | 0.15 | 0.73 | 0.35 | 0.07 | 96.92 | 0.70 | 0.11 | 0.21 | 1.66 | 88.85 |
| Red | 0.01 | 1.76 | 0.75 | 1.78 | 3.14 | 1.64 | 90.54 | 0.02 | 0.22 | 1.68 | 67.03 |
| Substratum | 0.67 | 0 | 0.14 | 0.03 | 0 | 0.22 | 0.25 | 96.59 | 0.15 | 51.78 | 99.90 |
| Water | 0.30 | 0 | 0.15 | 0.15 | 3.10 | 0.75 | 1.20 | 2.25 | 95.15 | 21.34 | 92.76 |
| Total | 100 | 100 | 100 | 100 | 100 | 100 | 100 | 100 | 100 | 100 | - |
| Prod. Acc. | 97.82 | 39.00 | 89.32 | 91.71 | 93.53 | 96.92 | 90.54 | 96.59 | 95.15 | - | - |

3.2.2. SAM Results

The SAM classifier, trained using image-derived spectra, revealed a similar cover of intertidal Fucales as MLC (27.6% of the site) (Figure 8). The overall classification accuracy for the SAM was 87.9% and the kappa coefficient was 0.82. By contrast with MLC, the four bathymetric levels appeared less distinct. The Pc-Fspi level was dominated by a thin band of both *P. canaliculata* and *F. spiralis* (1.4% and 1.3% of total pixels, respectively) with a better cover of *F. spiralis* than for MLC. The An and Fser levels (mid-shore) were dominated by a large band of *A. nodosum* and of *F. serratus* (5.5% and 3.5% of total pixels, respectively), but the cover of *F. serratus* was less important than for MLC. The He-Ld level (lower shore) was characterized by the important development of *H. elongata* (11.8% of total pixels). The cover of red macroalgae was distributed on all of the site (2.2% of total pixels) and was more present in the Fser level compared to the MLC results. Green algae were mainly present in the lower shore (1.9% of total pixels). The 'Substratum' and 'Water' classes represented the majority of the site (54.8% and 17.5% of the site, respectively). The macroalgal classes '*H. elongata*' and 'Green' showed the highest producer/user accuracies (Table 5). As for MLC, there was some misclassification. First, 18% of '*P. canaliculata*' pixels had been classified as '*F. spiralis*' (9.44%) and '*H. elongata*' (9.49%). The lowest producer/user accuracy was for '*F. spiralis*', with the largest misclassification (37.54%) in '*A. nodosum*' and 14.37% of pixels in '*P. canaliculata*'. Of the '*A. nodosum*' pixels, 18.76% were misclassified as '*F. spiralis*', but also 7.35% and 4.62% of '*A. nodosum*' pixels were misclassified as '*F. serratus*' and '*P. canaliculata*', respectively. Of the '*F. serratus*' pixels, 18.86% were misclassified as 'Red', and 17.37% of pixels should have been classified as '*F. serratus*', when they were in fact classified as '*A. nodosum*'. 'Green' and 'Red' algae were globally well classified, but there was some misclassification between 'Red' algae and some Fucales, such as '*A. nodosum*'. 'Substratum' and 'Water' classes had the highest producer/user accuracy and so were well classified on the entire image.

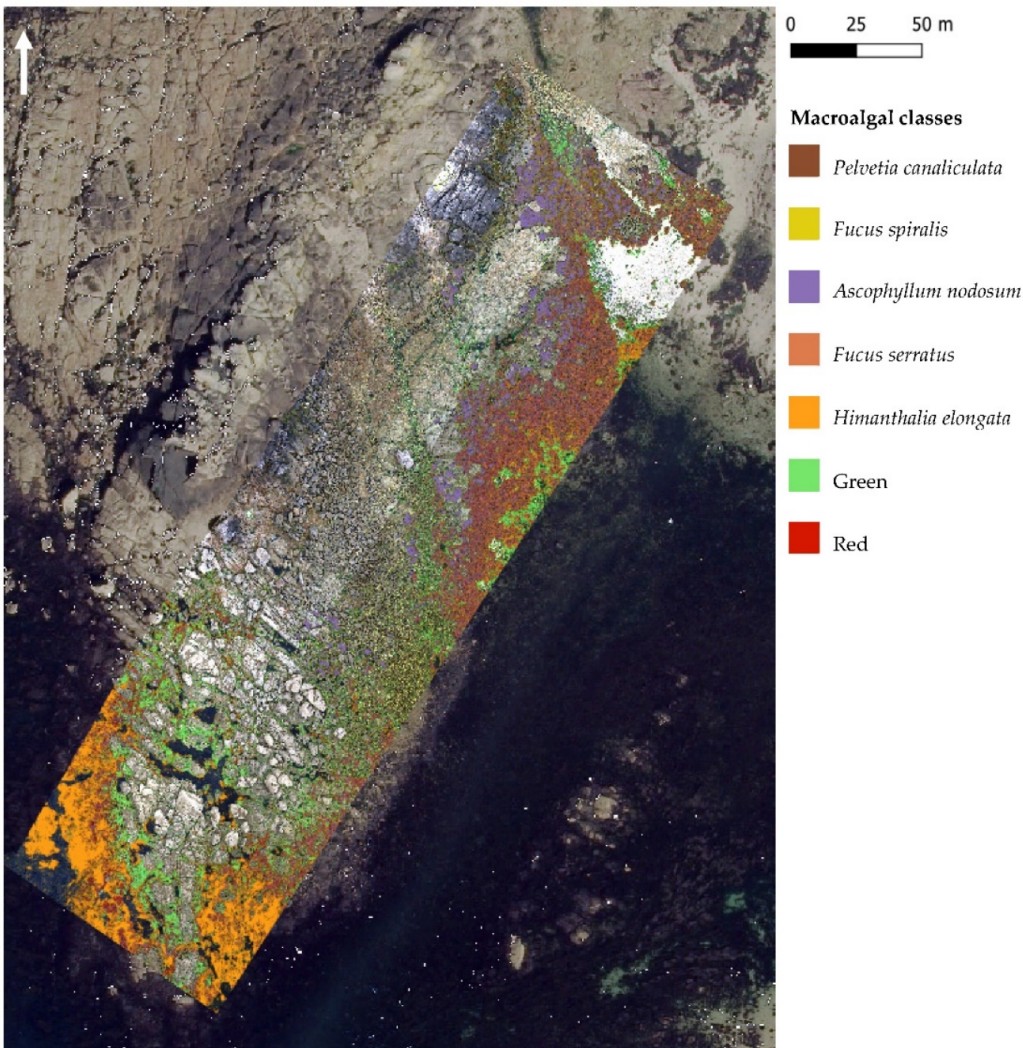

**Figure 8.** Spectral angle mapper (SAM), trained using image-derived spectra resulting from the hyperspectral UAV survey at Porspoder. Seven macroalgal cover classes are displayed over the UAV RGB imagery and an orthophotography (Mégalis Bretagne et collectivités territoriales bretonnes—2015), giving an overview of the site. The 'Substratum' and 'Water' classes are not represented on the map. Class codes 'Green' and 'Red' represent grouped green and red macroalgal species, respectively.

**Table 5.** Spectral angle mapper (SAM) confusion matrix, calculated, using ENVI 5.6.1, by comparing pixels of known class locations to those predicted by the classification workflow, for each of the nine cover classes. Results are displayed as percentages of pixels assigned, correctly or incorrectly, to each class. User/producer accuracies (User Acc. and Prod. Acc., respectively) are also presented.

| Class | *P. canaliculata* | *F. spiralis* | *A. nodosum* | *F. serratus* | *H. elongata* | Green | Red | Substratum | Water | Total | User Acc. |
|---|---|---|---|---|---|---|---|---|---|---|---|
| Unclassified | 0 | 0 | 0 | 0 | 0.06 | 0 | 0 | 0 | 0.24 | 0.06 | - |
| *P. canaliculata* | 65.22 | 14.37 | 4.62 | 9.604 | 0.72 | 6.32 | 3.97 | 0.03 | 0.04 | 1.44 | 15.44 |
| *F. spiralis* | 9.44 | 25.95 | 18.76 | 2.78 | 0 | 0.33 | 0.70 | 0.01 | 0 | 1.35 | 0.43 |
| *A. nodosum* | 3.02 | 37.54 | 67.06 | 17.37 | 0.04 | 0.32 | 22.84 | 0.01 | 0.01 | 5.48 | 70.97 |
| *F. serratus* | 1.97 | 8.80 | 7.35 | 38.80 | 1.36 | 1.44 | 8.21 | 0.01 | 0.04 | 3.54 | 80.26 |
| *H. elongata* | 9.49 | 1.47 | 0.44 | 11.78 | 95.01 | 10.76 | 7.26 | 0.01 | 7.92 | 11.76 | 75.96 |
| Green | 7.72 | 6.45 | 0.77 | 0.75 | 0.86 | 77.33 | 1.42 | 0.64 | 0.73 | 1.89 | 62.07 |
| Red | 2.56 | 5.13 | 0.93 | 18.86 | 0.60 | 0.23 | 54.82 | 0 | 0.02 | 2.19 | 31.09 |
| Substratum | 0.29 | 0 | 0.05 | 0.01 | 0 | 0.43 | 0.06 | 99.04 | 8.41 | 54.81 | 96.79 |
| Water | 0.30 | 0.29 | 0.02 | 0.02 | 1.35 | 2.82 | 0.72 | 0.24 | 82.60 | 17.49 | 98.23 |
| Total | 100 | 100 | 100 | 100 | 100 | 100 | 100 | 100 | 100 | 100 | - |
| Prod. Acc. | 65.22 | 25.95 | 67.06 | 38.8 | 95.01 | 77.33 | 54.82 | 99.04 | 82.6 | - | - |

### 3.3. Comparison of Field Sampling and Hyperspectral Classification

Covers determined by field sampling are compared here to the classification results by both MLC and SAM. A visual representation of the comparison between in situ sampling, an infrared picture and the two methods is provided in Figure 9 and Appendix A Figures A1–A3.

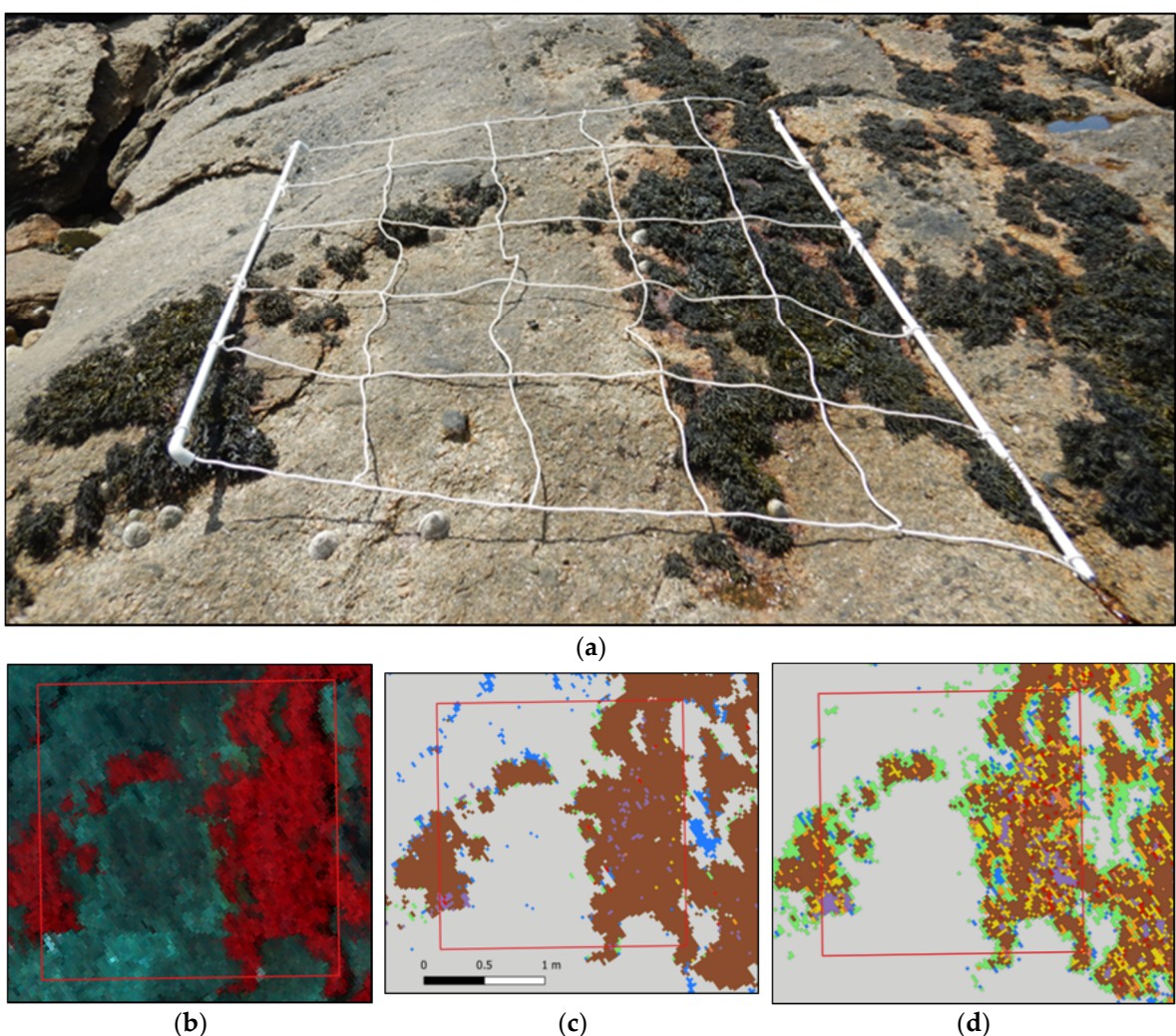

**Figure 9.** (**a**) Picture of a sampling spot on the Pc-Fspi level at Porspoder taken during field sampling in June 2021. (**b**) NIR-G-B image of the same sampling spot. (**c**) Result of the MLC classification. (**d**) Result of the SAM classification. The red square corresponds to the mobile grid structure used for field sampling. Color code in (**c**,**d**) corresponds to the following classes: '*P. canaliculata*' (brown), '*F. spiralis*' (yellow), '*A. nodosum*' (purple), '*F. serratus*' (coral), '*H. elongata*' (orange), 'Red' algae (red), 'Green' algae (green), 'Substratum' (grey) and 'Water' (blue).

To compare covers estimated in situ to those obtained through hyperspectral classification, in situ erect and crustose red algae on one side, and *H. elongata* and *L. digitata* on the other side, have been mixed up in order to attain classes similar to remote classification (Figures 10 and 11).

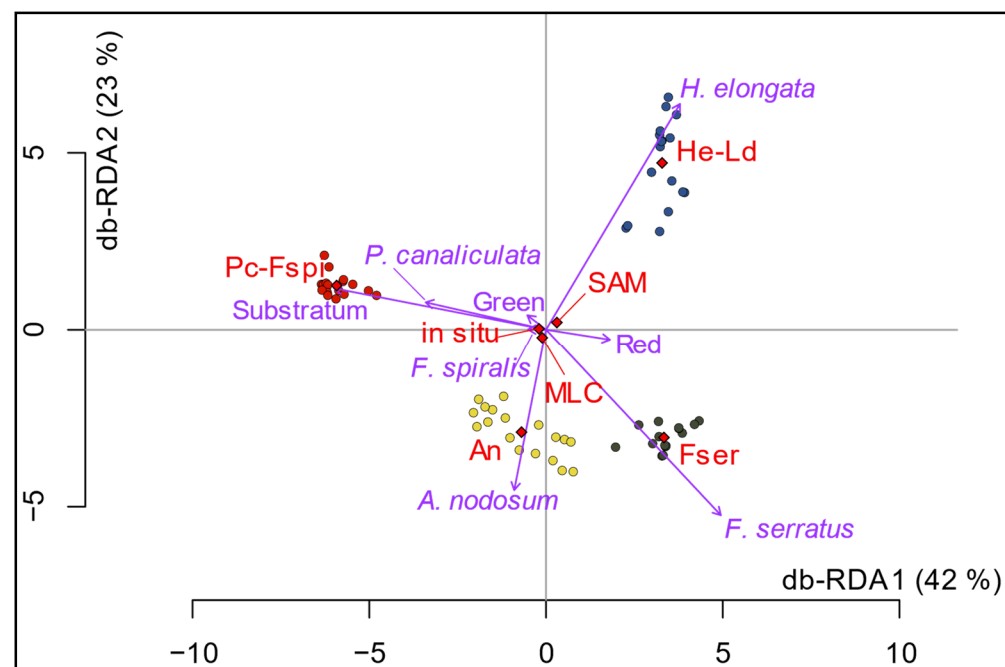

**Figure 10.** Db-RDA (scaling 1) performed from the macroalgal and pixels covers data for the in situ and the two classifications algorithms. Sampling spots are indicated by circles (Pc-Fspi = red circles, An = yellow circles, Fser = black circles and He-Ld = blue circles), variables (classes used in situ and for classifications: *Pelvetia canaliculata*, *Fucus spiralis*, *Ascophyllum nodosum*, *Fucus serratus*, *Himanthalia elongata*, Red, Green and Substratum) appear in purple, and the explanatory variables (level: Pc-Fspi, An, Fser and He-Ld; method: in situ, MLC and SAM) in red.

The results of the db-RDA are shown in Figure 10. Only the distance between objects (replicates) is considered; therefore, Scaling 1 has been chosen. The model is significant, with an $R^2$ of 91% (F-value of 117.89, $p$-value $< 0.001$). Axes 1 and 2 explain a significant portion of the total variability, with 42% and 23%, respectively. The points, corresponding to the replicates, are grouped into four homogeneous clusters, in agreement to the four bathymetric levels, with no significant distinction between the three methods. The explanatory variable 'level' is obviously correlated to the four clusters, with 'Pc-Fspi', 'An', 'Fser' and 'He-Ld' associated with the clusters corresponding to these four levels while the variable 'method' appears to have no significant influence on the distribution of replicates within each level. The results obtained from the three methods thus appear similar.

When looking in detail at the classes, the methods appeared broadly similar (Figure 11).

In Pc-Fspi (Figure 11a), the cover of the dominating Fucales '*P. canaliculata*' estimated by the in situ sampling was 32.5%, showing no significant difference to those obtained from MLC (39.9%) and SAM (20.9%) (Kruskal–Wallis, $p > 0.05$). On the contrary, for the other dominating Fucales '*F. spiralis*', the test showed no significant difference between in situ sampling (5.0%) and both methods, but a significant difference was observed between the two models (0.6% for MLC and 5.5% for SAM, Kruskal–Wallis, $p < 0.04$). In contrast, the cover of 'Red' macroalgae was 5.0% in situ, showing a single significant difference with the method MLC (0.2%, Kruskal–Wallis, $p < 0.001$). The estimation performed by SAM (1.4%) did not show any significant difference with the other two methods. Bare rock represented about half of the surface of sampling spots in this level, whatever the method considered, showing no significant differences (Kruskal–Wallis, $p > 0.05$).

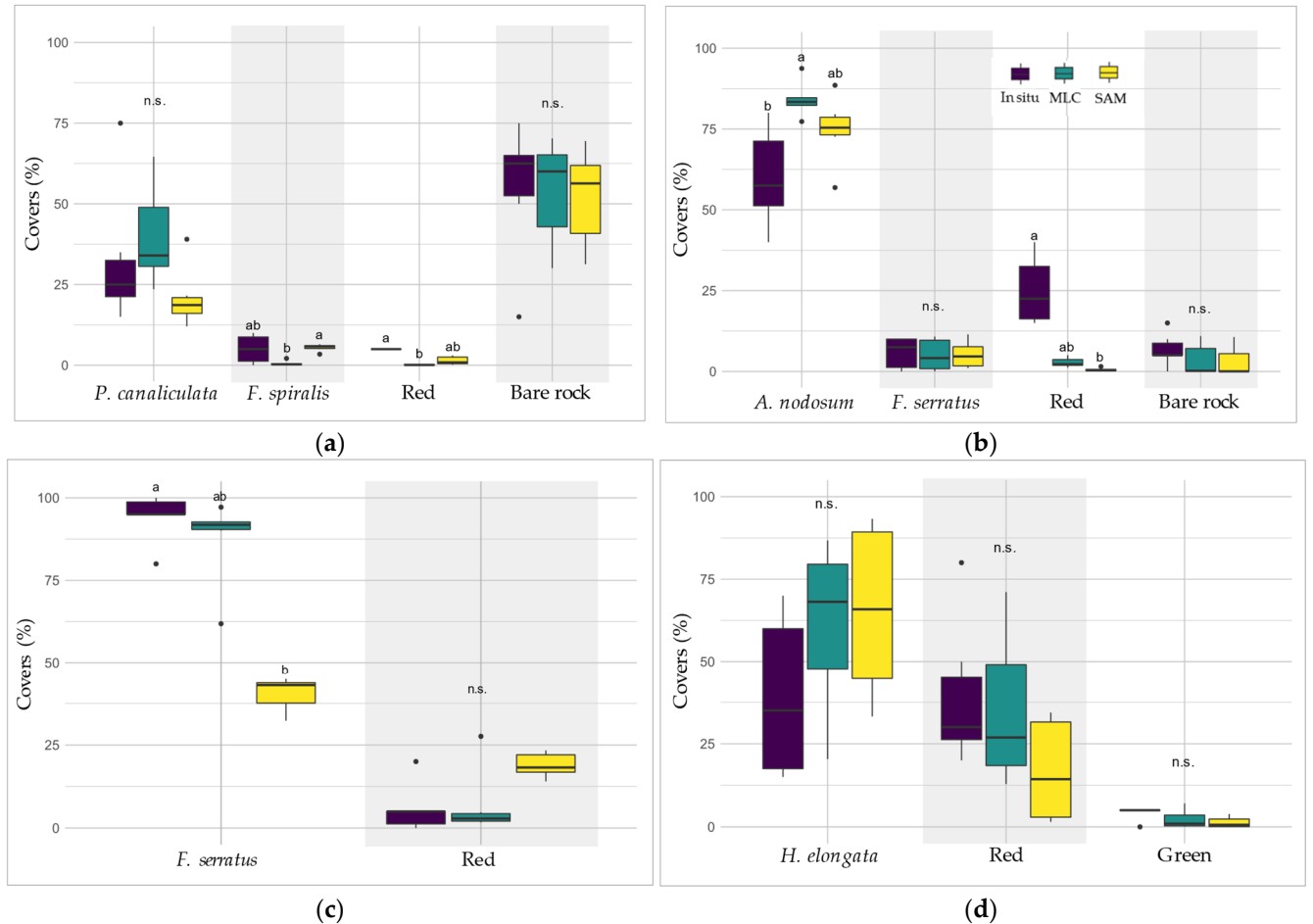

**Figure 11.** (**a**) Comparison of in situ (violet boxes), MLC (green boxes) and SAM (yellow boxes) determined covers of main Fucales species, macroalgal classes and bare rock found in Pc-Fspi (**a**), An (**b**), Fser (**c**) and He-Ld (**d**) levels. Covers are given in percentages. In the boxplots, only the classes showing covers for all three methods are represented. Letters refer to statistical differences (Kruskal–Wallis).

In An (Figure 11b), the cover of '*A. nodosum*' for SAM and MLC was higher (74.7% and 84.1%, respectively) than the in situ cover (60%). However, there was no significant difference in its covers between in situ and SAM, but a significant difference appeared between in situ and MLC (Kruskal–Wallis, $p < 0.05$). The '*F. serratus*' cover showed no significant difference between the three methods, i.e., circa 5% (Kruskal–Wallis, $p > 0.05$). The percentage of 'Substratum' was also close and did not show significant differences, with 6% in situ and ca. 3% for MLC/SAM (Kruskal–Wallis, $p > 0.05$). However, covers of 'Red' were significantly higher in the in situ field sampling (25%) than in the MLC method (Kruskal–Wallis, $p < 0.005$). The SAM method was not different for either of the two.

In the Fser level (Figure 11c), the cover of '*F. serratus*' was significantly greater in situ, with 94.2%, than in SAM with an estimated cover of 40.5% (Kruskal–Wallis, $p < 0.05$). MLC method did not show significant differences with either method (87.6%).

In He-Ld (Figure 11d), the '*H. elongata*' covers were 56.7% in situ, 61.3% with MLC and 65.5% with SAM and were statistically the same (Kruskal–Wallis, $p > 0.05$). 'Red' covers showed nearly similar values (39.2% and 34.9%, respectively). Even though SAM 'Red' covers were lower (16.8%), no significant difference was observed between the three methods (Kruskal–Wallis, $p > 0.05$). 'Green' covers were also not significantly different between in situ (4.2%), MLC (2.3%) and SAM (1.3%) (Kruskal–Wallis, $p > 0.05$).

To summarize these results, no generalization can be applied, both methods showed contrasted results according to the level and/or class considered. For instance, the cover of 'Red' is better estimated by the SAM in Pc-Fspi, and, on the contrary, better estimated by the MLC in An. The same observation can be made for the dominating Fucales, which are better represented by the SAM method in An, and by the MLC in Fser.

## 4. Discussion

Seaweeds are ecosystem engineers and key habitat-formers in temperate marine coastal ecosystems [73,74]. The use of satellite sensors to monitor populations of large temperate macroalgal species is well documented [54,75,76]. Such techniques have been used mainly to characterize the extension of macroalgal communities and related habitats [37]. However, the accurate classification of closely related macroalgal species (e.g., different species within the same Genus) and groups of species in remote sensing analysis remain a key point that still needs to be investigated. Using a high spatial and spectral resolution technology could be part of the solution [44].

In this study, heterogeneous seaweed habitats vertically distributed were successfully differentiated using both field and remote techniques. The 'undisturbed' sampling method [59] was used in situ to describe the structure of macroalgal communities and was then directly compared with the remote sensing imagery.

### 4.1. Habitats Characterization through Remote Sensing and Field Sampling

Orthophotos are often sufficient to remotely describe a habitat dominated by a single species, forming homogeneous populations, such as mussels [77,78] or polychaete reefs [79] that can be identified on a large scale. Eventually, species groups can be discriminated (brown, red and green seaweeds) [43], but this differentiation is quickly overtaken when studying complex ecosystems, showing an intricate microtopography, such as the European rocky shores. In the present study, using hyperspectral imagery, seaweed habitats were successfully differentiated (1) between them, (2) from substratum and (3) from seawater; the spectral signature allowed a clear differentiation between them (Figure 6a) as already reported in previous studies [13,42,75]. Moreover, the spectral signatures of five Fucales species were also differentiated, allowing an accurate mapping of the study site and its habitats. Here, two species of Fucus (*F. serratus* and *F. spiralis*), previously often gathered as a single class [35,44,58], were also discriminated. Indeed, the rocky shore surveyed during the present study showed a succession of several dominant fucoid species, with a conspicuous increase of red seaweeds abundance in the low intertidal zone, as already reported from previous studies in the area, using only field sampling [80,81]. However, technical limitations were spotted: limpets and barnacles were impossible to discriminate (although large limpets can be guessed) because of their heterogeneous distribution on rocky shore and close spectral signature with substratum. The same issue was observed for distinguishing red and green seaweed species, which were heterogeneously distributed and can present similar spectra, or variations of spectra according to their health conditions (e.g., pigment degradation, grazing, occurrence of epi/endophytes) [42,82,83].

In this study, we assigned several subclasses in the 'Red' one because of the dominance in He-Ld by an assemblage of *Mastocarpus stellatus/Chondrus crispus*. That assemblage masked the occurrence of several filamentous or turf red algae which were not identified on hyperspectral images, but with the use of existing spectral libraries [82], it may be an option [44]. In the same way, crustose and erect red macroalgae were also assigned in the 'Red' class for the same reasons, but with distinguishable patches, it could be possible to create two different classes [42]. For further analysis, it would be interesting to refine classes and to include more identified red species to the classification. The kelp *Laminaria digitata* was not added to the classification due to residual seawater on the images. However, it would be useful to separate that species as a spectral class due to its dominancy in the upper sublittoral zone [44,58]. It would also be interesting to transfer the spectral library created for other study sites to check if the method is interoperable.

### 4.2. Comparison of the Two Classifiers

Our results showed that remote classification data were in agreement with covers calculated from field sampling inside sampling spots, in spite of an approximation of 5% for in situ estimations, and whatever spectral properties of brown macroalgae, which are very similar [82,84] (Figure 6b).

MLC provided the best producer/user accuracies for dominating algae of three levels (An, Fser and He-Ld) (Table 4) and SAM for two levels (An and He-Ld) (Table 5). Pc-Fspi provided the lowest producer/user accuracy, with the lowest values for '*F. spiralis*' for the two classifications. This could be a site effect due to the lower extension of these two species compare to the others in Porspoder, especially *F. spiralis*, which results in a reduced pixels selection for the algorithms. Moreover, the fact that *F. spiralis* is found in more shaded environments/crevices on this site could affect the data since shadows cause serious difficulties for remote shooting [85]. There is also a clear misclassification between *P. canaliculata* and *F. spiralis,* especially for MLC, which might be explained because of forming confused communities with small size species, and similar colors, in this site.

In An, MLC was less accurate than SAM for *A. nodosum*. Due to the presence of the red alga *Vertebrata lanosa*, which was not taken into account for the classifications, the '*A. nodosum*' class was not considered as a 'pure class' and SAM had confusion overestimating 'Red' seaweed in An and Fser levels. There was also a little confusion in '*A. nodosum*' with *P. canaliculata,* as in Rossiter et al. (2020) [58]. This confusion appeared in the upper limit of *A. nodosum* distribution, where *A. nodosum* appeared brighter due to a stronger light stress [86], with a color close to that of *P. canaliculata*. Not surprisingly, *P. canaliculata* was also classified as *A. nodosum* when it was darker than usual.

The 'Red' class had a lower producer/user accuracy for the SAM classification, with an overestimation in Fser to the detriment of *F. serratus*, which is underestimated (Table 5). For that level, SAM is not a good descriptor at the site level.

On the processed images (Figures 7 and 8), many pixels were classified as 'Green', despite a reduced cover of green seaweeds in the field data. This is mainly due to the positioning of the spots on the shore, chosen because of a clear dominance by brown macroalgae.

Both SAM and MLC misclassified pixels of *H. elongata,* with some occurrence in Pc-Fspi, An and Fser, whereas this species does not develop in higher intertidal zones [7,87]. On the contrary, the high-level species *F. spiralis* could appear in An with both MLC and SAM. An approach taking into account the bathymetric range (strongly affecting certain species) could be determined using a lidar approach [88] and could solve such a problem, considering the vertical zonation of species [17].

The MLC classification was found to be more accurate than SAM at the site level due to better management of spatial heterogeneity of habitats by the MLC, and because SAM does not consider the magnitude of pixels' vectors. Moreover, groups of macroalgal species, Fucales in particular, have close spectral similarities, which could partially explain the lower accuracy of SAM [47,89]. Nevertheless, both supervised classifications mapped the four bathymetric levels sampled in this study and both classifiers were able to separate brown, red and green algae. At sampling spots scale, they provided similar results, so, they can be used to compare macroalgal covers with accuracy. To refine the results, it could be interesting to test an object-based classification that takes into account not only spectral information but also the shape, size, texture, tone and the compactness [90,91] of objects. In that prospect, macroalgae could be an interesting model due to various morphologies and textures.

### 4.3. Consistency of Specific Identification and Perspectives

Automated macroalgal classification applied to shores dominated by a single fucoid species is currently manageable [43]. However, discriminating and mapping *Fucus* spp. remain a challenge, as seen in previous studies, in which *Fucus* spp. have been gathered in a single mixed class [35,44,58]. Overall accuracy, used to estimate the quality of the

classification [62,92], presented values indicating a clear distinction between *F. spiralis* and *F. serratus*. Even though the final classification of the entire site of Porspoder pointed out some pixels which were not correctly attributed, the entire distribution of pixels on the site was consistent. The analysis by db-RDA did not show a significant influence by the method in the distribution of the sampling spots, unlike the level. This may be related to the nature of the site studied, with the Pc-Fspi level having much lower macroalgal covers than the An and Fser levels, and to the spectral properties of each species. Moreover, the comparison between in situ covers with MLC and SAM data showed no significant difference for most classes (Figure 11), therefore, validating the algorithms as good descriptors for intertidal macroalgal covers at sampling spots scale. This also confirm the db-RDA results; efficiency does not depend on the method but on the studied bathymetric level. However, there were more differences (at the site scale) with SAM which resulted in more misclassified pixels on images (Figures 9 and A1–A3). So, the use of common algorithms could be perfectible, but it does answer the problem of the study by classifying correctly macroalgal communities. Other algorithms such as random forests or support vector machines might be considered to estimate entire shores, as for coastal/terrestrial objects [93–97].

Indeed, the routine use of the hyperspectral method could be the subject of a long-term study in an ecosystem monitoring context, particularly in the context of Fucales regression on European coasts [98,99]. Indeed, since the last century, covers of some Fucales species have decreased under the action of various factors such as the intensification of grazing [100]. This trend is also well known and studied in various marine phanerogam species [101], and also in kelp species, submitted to increasing grazing and/or heat waves related to global change [102–104]. Indeed, hyperspectral imagery is already being used in many ecosystems in the context of conservation biology [105,106]. Thus, the promising results obtained in this work could serve as a basis for a conservation/monitoring program of intertidal habitats.

## 5. Conclusions

In light of the results, MLC seems to be a better classifier for mapping a seaweed-dominated rocky shore, with a more realistic achievement. To better assess the impact of global change on coastal ecosystems, there is an increasing interest in remote sensing data to evaluate the ecological state of corresponding habitats [107]. Otherwise, the community approach in ecological surveys gives a good opportunity to better understand functional traits of marine vegetation, including relationships with primary production [108,109]. In that context, this study gives for the first time a comparison of cover data for macroalgal habitats obtained by both in situ sampling on the shore and hyperspectral imagery at a centimeter resolution, and a consistent cartography of a site using well-known algorithms.

Our results go beyond the global distribution of macroalgal covers as inferred from indices such as NDVI, VCI or IP [35,110,111], but rather provide information on the fine scale repartition of species/groups of species on the shore.

Since coastal rocky shores integrate various and imbricated habitats, the UAVs approach developed here seems to be an adequate tool to evaluate the distribution of macroalgal communities/habitats at the site to geographical area level. Moreover, hyperspectral imaging at the centimeter scale allows for a precise analysis of the seaweed habitat structure in parallel to field monitoring.

**Author Contributions:** Collected field sampling data, methodology, analyzed and interpreted field and remote sensing data, wrote the article, W.D.; remote sensing methodology, made critical revisions for important intellectual content, A.L.B., T.B. (Touria Bajjouk) and S.R.; data analysis, made critical revisions for important intellectual content, M.H.; collected field sampling data, data analysis, made critical revisions for important intellectual content, T.B. (Thomas Burel); hyperspectral acquisition and pre-treatment, made critical revisions for important intellectual content, M.L. and A.G.; conceived and designed the study, supervision, collected field sampling data, funding acquisition, made critical revisions for important intellectual content, E.A.G. All authors have read and agreed to the published version of the manuscript.

**Funding:** This study was founded by the DREAL (Direction régionale de l'environnement, de l'aménagement et du logement) and the Regional Council of Brittany (Rebent Network), the Office Français de la Biodiversité and the Agence de l'Eau Loire-Bretagne (European Water Framework Directive and Marine Strategy Framework Directive). Wendy Diruit received a fellowship from the Doctoral School of Marine Sciences (Ecole Doctorale des Sciences de la Mer et du Littoral).

**Data Availability Statement:** The data presented in this study are available on request from the corresponding author.

**Acknowledgments:** Authors thank Sara Terrin and Alain Guenneguez for helping with the field sampling, and Marion Jaud for her valuable advice in hyperspectral analysis and the use of ENVI.

**Conflicts of Interest:** The authors declare no conflict of interest.

**Appendix A**

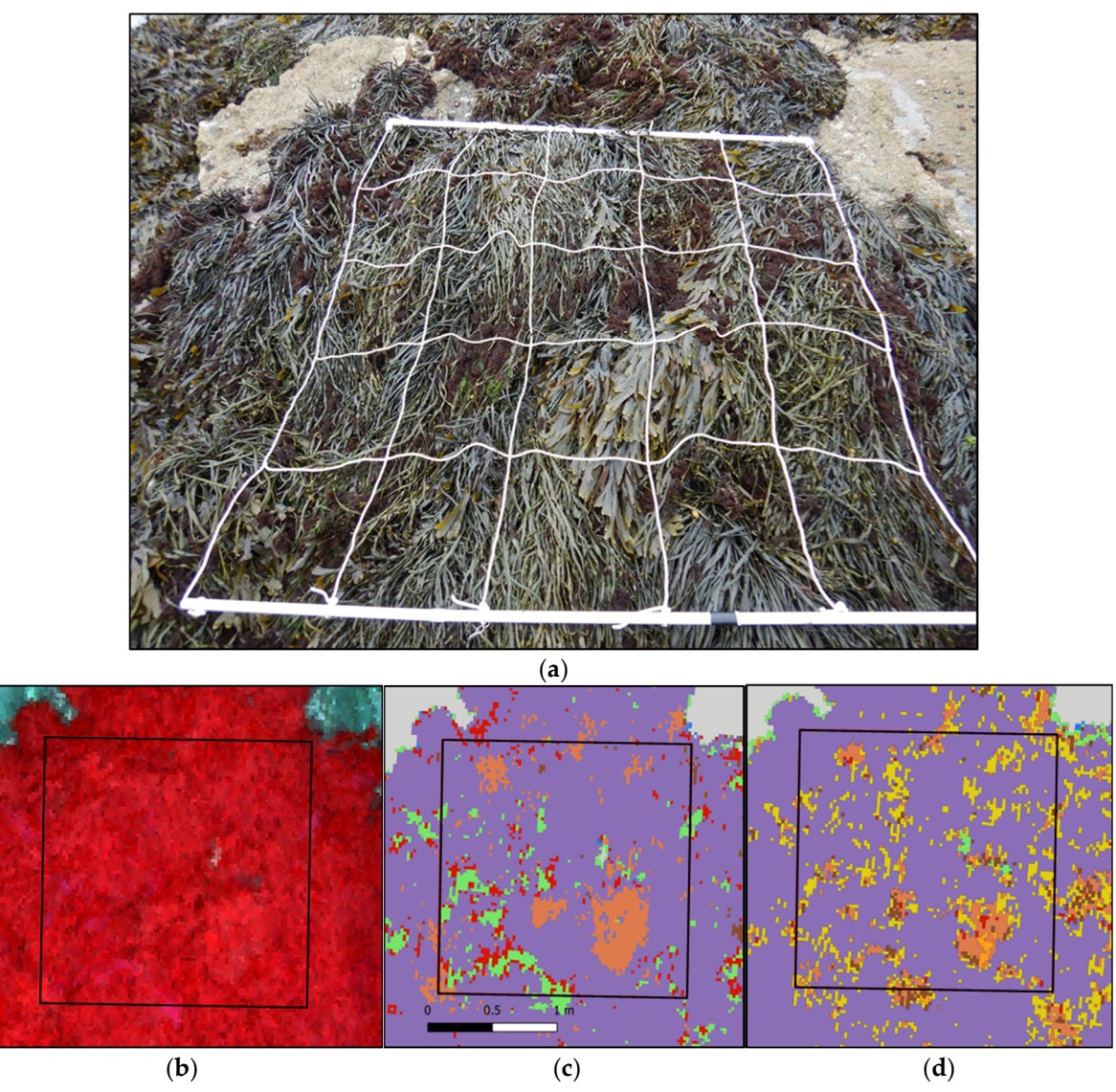

**Figure A1.** (**a**) Picture of a sampling spot on the An level at Porspoder taken during field sampling in May 2021. (**b**) NIR-G-B image of the same sampling spot. (**c**) Result of the MLC classification. (**d**) Result of the SAM classification. The black square corresponds to the mobile grid structure used for field sampling. Color code in (**c**,**d**) corresponds to the following classes: '*P. canaliculata*' (brown), '*F. spiralis*' (yellow), '*A. nodosum*' (purple), '*F. serratus*' (Coral), '*H. elongata*' (orange), 'Red' algae (red), 'Green' algae (green), 'Substratum' (grey) and 'Water' (blue).

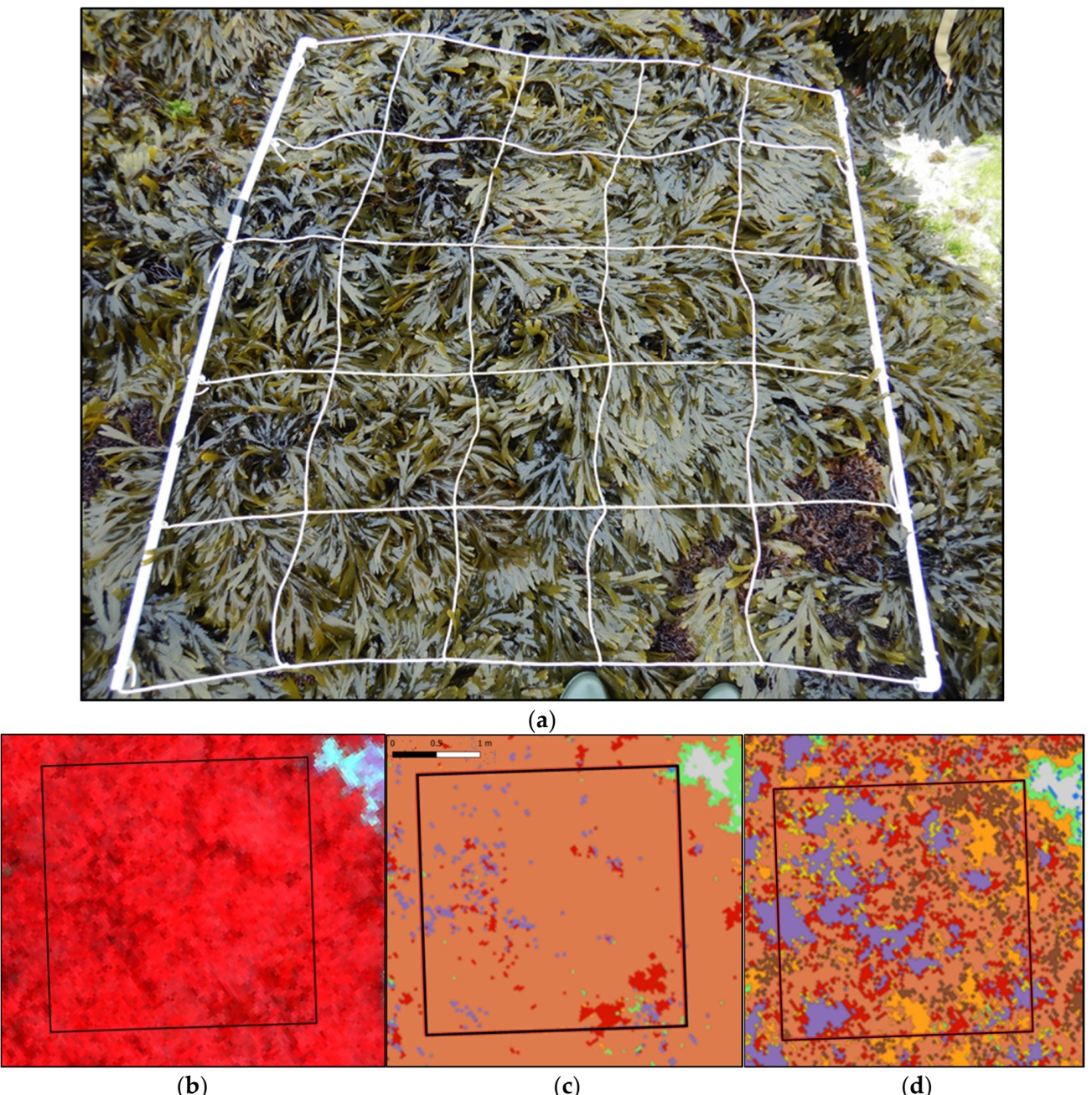

**Figure A2.** (**a**) Picture of a sampling spot on the Fser level at Porspoder taken during field sampling in June 2021. (**b**) NIR-G-B image of the same sampling spot. (**c**) Result of the MLC classification. (**d**) Result of the SAM classification. The black square corresponds to the mobile grid structure used for field sampling. Color code in (**c**,**d**) corresponds to the following classes: '*P. canaliculata*' (brown), '*F. spiralis*' (yellow), '*A. nodosum*' (purple), '*F. serratus*' (Coral), '*H. elongata*' (orange), 'Red' algae (red), 'Green' algae (green), 'Substratum' (grey) and 'Water' (blue).

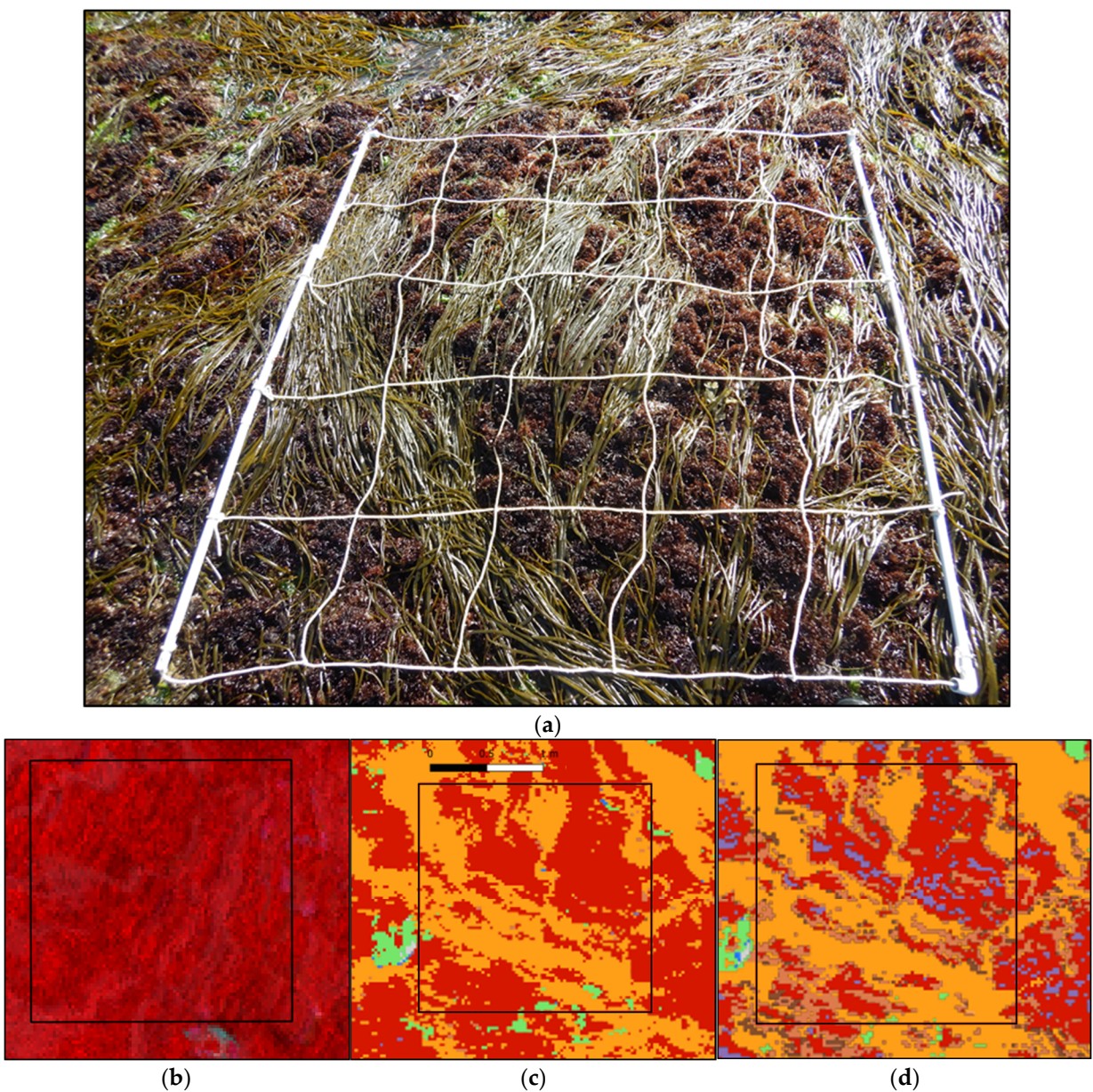

**Figure A3.** (**a**) Picture of a sampling spot on the He-Ld level at Porspoder taken during field sampling in April 2021. (**b**) NIR-G-B image of the same sampling spot. (**c**) Result of the MLC classification. (**d**) Result of the SAM classification. The black square corresponds to the mobile grid structure used for field sampling. Color code in (**c**,**d**) corresponds to the following classes: '*P. canaliculata*' (brown), '*A. nodosum*' (purple), '*F. serratus*' (Coral), '*H. elongata*' (orange), 'Red' algae (red), 'Green' algae (green), 'Substratum' (grey) and 'Water' (blue).

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
