# Peer review of "Seaweed Habitats on the Shore: Characterization through Hyperspectral UAV Imagery and Field Sampling"

_remotesensing, doi:10.3390/rs14133124_

Round 1

Reviewer 1 Report

The paper with title “Distribution of seaweed habitats on the shore by combining field sampling and hyperspectral imagery from UAV” has been reviewed. The article aims to evaluate the effectiveness of classification methods such as MLC and SAM in the study of algae through hyperspectral images acquired by drone.

The manuscript is clear and can be followed easily. Please find comments and critiques regarding the article's content in the following list:

  1. The title suggests that data acquired by UAV and field samplings are combined, however this is not true because field samplings are used as training and testing for classification methods. For this reason the title should be changed.
  2. The abstract and introduction are well written, however the aim of this study is not clearly stated.
  3. Regarding the classification methods adopted, I am not sure of the results obtained. Generally, it is good practice that when training sites are taken, they must contain quite the same number of pixels, for this reason I ask the authors to report the number of pixels examined for each class. It seems to me that there is no balance between the classes, so I think either that the trainings were not taken homogeneously or that we tried to get too many classes. It is also quite strange that the results obtained with MLC and SAM are so different from each other, especially the inversion of values between F. Serratus and F. Spiralis surprises me. I am sorry to say that such low User Accuracy values are not presentable in a scientific article unless accompanied by a plausible explanation. In particular, 6.23% and 0.43% are values that attest to the lack of effectiveness of the classification, but also values such as 26.04% relating to the User Accuracy of P. canaliculata are still excessively low.
  4. The references introduced are consistent, but too few of them concern the classification techniques, I advise the authors to add more bibliography on these techniques, in particular SAM and MLC.

The structure of the article is fine, the work carried out in the field is also noteworthy. However, the article does not report any scientific innovation, nor any other type of novelty. The methods applied are well known in the literature, but the results obtained need to be reviewed.

For these reasons I do not consider the manuscript suitable for publication in this journal, but in light of the field work that has been carried out, I encourage the authors to re-submit it after having carefully reviewed it (I am referring especially to the results).

Reviewer 2 Report

Please see PDF

Reviewer 3 Report

Please find my comments in the attached file
